# Poly(Lactic Acid)-Based Graft Copolymers: Syntheses Strategies and Improvement of Properties for Biomedical and Environmentally Friendly Applications: A Review

**DOI:** 10.3390/molecules27134135

**Published:** 2022-06-28

**Authors:** Jean Coudane, Hélène Van Den Berghe, Julia Mouton, Xavier Garric, Benjamin Nottelet

**Affiliations:** 1Department of Polymers for Health and Biomaterials, Institut des Biomolecules Max Mousseron, UMR 5247, University of Montpellier, CNRS, ENSCM, 34000 Montpellier, France; helene.van-den-berghe@umontpellier.fr (H.V.D.B.); xavier.garric@umontpellier.fr (X.G.); benjamin.nottelet@umontpellier.fr (B.N.); 2Polymers Composites and Hybrids, IMT Mines d’Alès, 30100 Alès, France; julia.mouton@epf.fr; 3EPF Graduate School of Engineering, 34000 Montpellier, France

**Keywords:** poly(lactic acid), chemical modification, graft copolymers, compatibilization, biomedical and environmental applications

## Abstract

As a potential replacement for petroleum-based plastics, biodegradable bio-based polymers such as poly(lactic acid) (PLA) have received much attention in recent years. PLA is a biodegradable polymer with major applications in packaging and medicine. Unfortunately, PLA is less flexible and has less impact resistance than petroleum-based plastics. To improve the mechanical properties of PLA, PLA-based blends are very often used, but the outcome does not meet expectations because of the non-compatibility of the polymer blends. From a chemical point of view, the use of graft copolymers as a compatibilizer with a PLA backbone bearing side chains is an interesting option for improving the compatibility of these blends, which remains challenging. This review article reports on the various graft copolymers based on a PLA backbone and their syntheses following two chemical strategies: the synthesis and polymerization of modified lactide or direct chemical post-polymerization modification of PLA. The main applications of these PLA graft copolymers in the environmental and biomedical fields are presented.

## 1. Introduction

Today, bioplastics, compounds derived from sustainable sources, are one of the best alternatives to petroleum-based plastics. They are natural or synthetic biopolymers, and include poly(lactic acid) (PLA), which is of great commercial interest due to various factors. First, PLA is produced by polymerizing lactide, a derivative of lactic acid industrially produced from plants, making it a biosourced thermoplastic. As such, extrusion, molding, injection molding, thermoforming, and fiber spinning are largely used to process PLA for many industrial applications. Lastly, it forms an intrinsically biocompatible system in a living environment and is biodegradable, with a tunable degradability as a function of its molecular weight and tacticity, which makes it suitable for many applications in biomedical and environmental fields such as tissue engineering, drug delivery, “green” packaging, textiles, etc. [1,2].

Despite these clear advantages, PLA suffers from some limitations. From an economical point of view, it remains more expensive than many non-biodegradable commodity polymers. Moreover, regarding its thermomechanical properties, it has a low toughness and poor impact strength. To overcome these limitations, polymer blends can provide the desired properties at a low cost through simple physical processes, rather than chemical approaches such as copolymerization reactions. Polymer blends and composites [3], as well as plasticizers [4], are used to improve the mechanical properties of polymers. The melt blending of dissimilar polymers is a classic method for obtaining new enhanced properties. Unfortunately, PLA-based blends exhibit an insufficient performance because the blended polymers are often thermodynamically immiscible, resulting in a poor compatibility between the blended components [5]. This phenomenon is particularly important in high-molecular-weight polymers commonly found in the field of orthopedics.

In order to circumvent the compatibility problem, compatibilizers have been proposed [6]. Compatibilizers are used to improve the properties of immiscible or partially miscible polymer blends. They improve the adhesion between the blended polymers. Compatibilizers can be “reactive” (they chemically react with at least one of the two blended polymers) or “non-reactive” (they have secondary interactions with both polymers). More generally, these PLA compatibilizers may consist of a copolymer comprising the PLA and the polymer to be compatibilized. PLA-based copolymers can be of a “block” or “graft” architecture. It is necessary to have at least one reactive function on the PLA to obtain these blocks or graft copolymers. However, PLA only has reactive functions at its chain ends, typically alcohol and carboxylic acid functions. Therefore, it is quite easy to prepare block copolymers (di-, tri- or multi-blocks) from PLA. The most common PLA-based block copolymers are probably the amphiphilic PLA-*b*-Poly(ethylene glycol) (PEG) di-block copolymers and PLA-*b*-PEG-*b*-PLA triblock copolymers, in which the hydrophobicity of PLA is decreased and its toughness is improved [7]. There are many review articles on the formation of PLA-based block copolymers and their applications, especially in the biomedical field [7,8,9].

The reactive functions at the chain ends of PLA can also react with the reactive functions in the chain of certain polymers, such as polysaccharides, to give polymer-*g*-PLA graft copolymers in a so-called “classic” structure (Figure 1), where the polymer backbone is grafted with PLA side chains [10]. The synthesis of “reverse” structures, i.e., with a PLA main chain grafted with other polymer side chains is more challenging because, unlike polysaccharides, the PLA backbone is not functionalized. Therefore, it is necessary to first functionalize the PLA chain before subsequent grafting of polymer segments onto the PLA backbone. From a theoretical point of view, two methods can be used to obtain a functionalized PLA backbone: (i) copolymerization of a lactide with a pre-functionalized lactide and (ii) direct chemical modification of a preformed PLA chain. It is therefore the aim of the present review to focus on these relatively uncommon reverse PLA-*g*-polymer structures.

## 2. Results

### 2.1. Copolymerization of Lactide with a Functionalized Lactide for the Preparation of PLA Graft Copolymers

To the best of our knowledge, according to the first method mentioned above, few functionalized lactides are described in the literature, and only a small proportion of these substituted lactides has been copolymerized with lactide to yield functionalized PLA backbones [11,12,13,14]. For example, some lactides functionalized with benzyl, allyl and propargyl groups have also been prepared and copolymerized with lactide, but no polymer chains were subsequently grafted onto the PLA backbone [15].

Yu et al. prepared an alkyne-functionalized lactide that was copolymerized with lactide to give an alkyne-functionalized PLA. An azide-paclitaxel-PEG was then reacted with the alkyne-functionalized PLA via a Cu(I)-catalyzed azide–alkyne cycloaddition (CuAAC) click reaction to give a novel graft polymer–drug conjugate (GPDC): PLA-*g*-Paclitaxel-PEG (Figure 2) targeting sustained release of Paclitaxel [16]. Zhang et al. prepared and (co)polymerized a dipropargyloxylactide by ring opening polymerization (ROP) in the presence of Sn(Oct)_2_ and TBBA (tertiobutylbenzyl alcohol). An azido-PEG (PEG-N3) was then grafted via CuAAC click chemistry (Figure 3) [17]. Turbidity, dynamic light scattering and NMR results suggested that these grafted copolymers exhibit a reversible thermo-responsive property, with LCST ranging from 22 to 69 °C depending on the molecular weight of the PEG.

In another synthetic method, Castillo et al. prepared a spirolactide–heptene monomer that yielded a PEG-grafted lactide after reaction with an azido-PEG (Figure 4) [18]. This PEG-grafted lactide was then polymerized with molecular weights above 10kDa. Typically, polymerization was carried out in anhydrous CH_2_Cl_2_ with 1,5,7-triazabicyclo [4.4.0]dec-5-ene (TBD)/benzyl alcohol as the catalyst/initiator system. Preliminary biological studies showed that PLA-*g*-PEG reduced non-specific protein adsorption and cell adhesion compared to the original PLA.

This limited number of examples illustrates that this approach based on functional-lactides is time-consuming, requires many steps and often gives low yields. For example, the global yields of the syntheses are 20% for dipropargyloxylactide [17] and 10% for PLA-*g*-Paclitaxel-PEG [16]. These copolymers are therefore expensive, and while this is not a fundamental problem for biomedical applications, it is a real drawback for environmental applications that use much larger amounts of product. Therefore, the second method, chemical modification of a preformed PLA chain, is the most widely used to obtain a functionalized PLA backbone allowing for the grafting of a second polymer.

### 2.2. Direct Functionalization of PLA Backbone: Towards PLA-Based Graft Copolymers

PLA is an aliphatic polyester known for being highly sensitive to chain breakings, especially in acid or alkaline environments. As a result, unlike poly ε-caprolactone, for which a general method of chain modification in anionic media is described [19], almost all substitution reactions on the PLA backbone are carried out under free radical conditions. In most cases, a functional small molecule is grafted on the PLA backbone to give it functionality and, in a second step, a polymer chain is grafted through this small molecule. This grafting can be carried out by polymerization of a monomer from this functional group (“grafting from”) or by the direct grafting of a polymer chain through its reactive chain end (“grafting onto”). From an experimental point of view, the functionalization is carried out in blenders or extruders in the presence of a radical precursor. Very frequently, the modification of the PLA chain aims to obtain a compatibilizing agent for blends of PLA and another polymer in order to improve the mechanical properties of the blends. Zeng et al. described the main basic strategies for the compatibilization of the PLA blends, among them functionalization of the PLA backbone with a reactive compound [20].

#### 2.2.1. PLA-*g*-Maleic Anhydride (PLA-*g*-MA)

The most widely used reagent for functionalizing the PLA chain is maleic anhydride (MA) due to its good chemical reactivity, low toxicity and good stability to the experimental synthesis conditions [21,22,23]. The grafting of MA onto the PLA backbone results in a reactive compatibilizer due to the presence of the anhydride function. PLA graft copolymers are therefore obtained by a reaction of the anhydride group with some reactive functions of the blended polymer.

Functionalization with MA is typically carried out between 120 °C and 200 °C with stirring at 50–200 rpm in an extruder in the presence of a radical initiators, such as dicumyl peroxide (DCP) or benzyl peroxide (BPO). A typical reaction scheme is shown in Figure 5 [21]. The role of different reaction parameters (MA concentration, nature of the initiator, temperature, molar mass of PLA) on the percentage of grafting is also described [23]. It was found that the molecular weights of PLA decreased during the reaction due to chain scissions. The grafting percentage of low-molecular-weight PLA is greater than that of the high-molecular-weight PLA because of steric hindrance (Table 1). It should be noted that the grafting percentage of MA increased with the initial concentration of MA, but remained low (<1.25%) regardless of the different parameters of the reaction, which shows the low efficiency of these radical reactions [23]. If one wishes to graft at least two molecules of anhydride per PLA chain, the molar mass of the latter must be higher than 14,500 g.mole^−1^ if substitution degree = 1%, which is not always the case. Nevertheless, the results obtained for the compatibility of various PLA-based blends are significant. Examples of variations in mechanical properties are shown in Table 2 in blends of PLA/cellulose nanofibers (CNF) [24].

The following paragraphs show the main PLA graft copolymers obtained from PLA-*g*-MA with their principal properties and applications.

##### PLA-*g*-Cellulosic Derivatives

Cellulose fibers derived from renewable biomass have attracted interest as microscale reinforcements in composite materials. Natural fibers have many advantages—low density, low cost, renewability, biodegradability—that make them excellent candidates for the design of biodegradable materials. They can advantageously replace mineral reinforcements in PLA matrices [3,4]. However, a poor compatibility between the fiber and the polymer matrix leads to materials with poor performances. In particular, nanocelluloses, due to their polar surfaces, are difficult to uniformly disperse in a non-polar medium. The consequences of this poor interfacial compatibility between polymer and filler are poor properties of the final blend. The compatibilization of PLA/cellulosic derivatives blends to improve many of the blends’ properties, especially mechanical properties, while maintaining a natural source, are the most widely described in the literature [24,25]. Because PLA-*g*-MA acts as a reactive compatibilizer, a PLA-*g*-cellulose copolymer is formed as a result of the reaction between the anhydride of PLA-*g*-MA, and alcohol functions of the cellulosic derivative (Figure 6) [21]. The presence of a very low percentage of PLA-*g*-MA (<1%) in the original blend significantly improved the properties of the blend [24]. Tensile strength, tensile modulus and strain at break were increased by 55.3%, 15.45% and 30.4%, respectively, over neat PLA by adding 5 wt.% of cellulose nanofibers (CNFs), and by 169.2%, 36.3% and 139.1%, respectively, by adding 5% of PLA-*g*-MA to the blend PLA/CNF.

Many cellulosic derivatives were introduced into PLA matrices in the presence of PLA-*g*-MA to improve the mechanical properties of blends such as Luffa [26], flax [27], coffee grounds [28], wood flour or rice husk [29,30], sisal fibers [31], straw [32], bamboo fiber [33], cassava starch [34,35], starch [36], and lemongrass fiber [37]. For example, the use of PLA-*g*-MA in a PLA/cassava starch blend has a significant impact on elongation at break but not on Young’s modulus or tensile strength. However, it was noted that PLA-*g*-MA with a higher proportion of grafted MA (0.52 wt.%) had a lower molecular weight and higher dispersity value, showing some degradation of the polymer backbone [34,38].

Many other properties are improved in PLA-*g*-MA compatible blends, such as morphological, rheological, thermal, tensile and moisture sorption properties as well as thermal degradation [39,40,41]. For example, Figure 7 shows SEM micrographs of PLA/TPS (thermoplastic starch) blends (70/30 *w*/*w*) without (Figure 7a) and with two parts per hundred rubber (phr) PLA-*g*-MA (Figure 7b), highlighting the compatibilization of the blend of PLA-*g*-starch formed in situ.

To obtain a PLA-*g*-starch copolymer, another method is to react maleic anhydride with starch to obtain a maleated thermoplastic starch (MTPS), which is then mixed with PLA in the presence of Luperox 101 (2,5-bis(tert-butylperoxy)-2,5-dimethylhexane) in a Brabender at 180 °C for 5 min. The reaction scheme is shown in Figure 8 [42].

Direct grafting of cellulose nanocrystals (CNC) on PLA, without the addition of PLA-*g*-MA, is also described, following the reaction scheme of Figure 9 [43]. could his DCP was sprayed onto PLA beads, and the DCP-coated PLA pellets were mixed with CNC and extruded in a twin-screw extruder at 180 °C at 50 rpm for 5 min. The effective grafting of CNC onto PLA was identified by SEC, FTIR and NMR, but NMR showed a very low proportion of CNC in the copolymer. Some mechanical and structural properties were significantly impacted (increased Young modulus, decreased elongation, increased crystallinity).

##### PLA-*g*-Natural Rubber (PLA-*g*-NR)

Among the drawbacks of PLA materials, we can also highlight their fragility. Natural rubbers (NR), on the other hand, are highly flexible, environmentally friendly and derived from a renewable resource. They are good toughness agents due to their high molecular weight and very low glass transition temperature. However, due to the non-polarity of NRs, PLA/NR blends are immiscible and not compatible. To improve the interfacial interaction between PLA and NR, the reactive compatibilizer PLA-*g*-MA is used to form a graft copolymer PLA-*g*-NR [44,45]. Typically, the compatibilized blend is made in a twin screw extruder at a temperature between 160 and 180 °C and a screw speed of around 30 rpm. With the addition of PLA-*g*-MA, the mechanical properties of the material were significantly improved. It was found that a 3% PLA-*g*-MA was the best compatibilizer composition to achieve the best performance of the material [44]. The reverse reaction of a maleic anhydride on NR (NR-MA), followed by reaction on PLA in a radical medium, was also performed with similar results regarding mechanical properties [46,47]. However, in this case, the proposed mechanism does not involve a reaction on the PLA backbone but only the alcohol chain end, which reacts on the backbone of NR-MA (Figure 10).

##### PLA-*g*-Polyester

Other degradable and flexible polyesters are now being developed on an industrial scale, such as polybutylene adipate-co-terephthalate (PBAT), polybutylene succinate (PBS) or poly ε−caprolactone (PCL); the first two examples are derived from renewable resources. Blends of PLA with these polyesters are good candidates for improving the brittleness and toughness of PLA while maintaining the biodegradability of the blends. Unfortunately, these blends are largely incompatible, especially with PBAT, due to its structure, which contains aromatic rings [48]. As with PLA/NR blends, PLA-*g*-MA was used as a reactive compatibilizer whose possible reaction of the alcohol chain end of PBAT with the grafted anhydride is shown in Figure 11.

Typically, the reaction is carried out in a co-rotating twin screw extruder at a temperature between 160 and 180 °C and a screw speed of 25–100 rpm [49,50]. Effective grafting of PBAT was demonstrated by a drastic increase in the molecular mass of the copolymer, from 10 kDa to ca. 20 kDa. The mechanical properties of the PLA/PBAT blend were only slightly improved despite the incorporation of PLA-*g*-MA. However, the improvement of the interfacial adhesion between PLA and PBAT was evidenced by SEM micrographs. A very limited compatibility effect of the addition of PLA-*g*-MA in PLA/PBAT/TiO_2_ blends was also observed, with TiO_2_ acting as a nucleating agent [51]. The addition of CaCO_3_ to PLA/PBAT/PLA-*g*-MA blends resulted in an increase in Young’s modulus [52].

**Figure 11 molecules-27-04135-f011:**
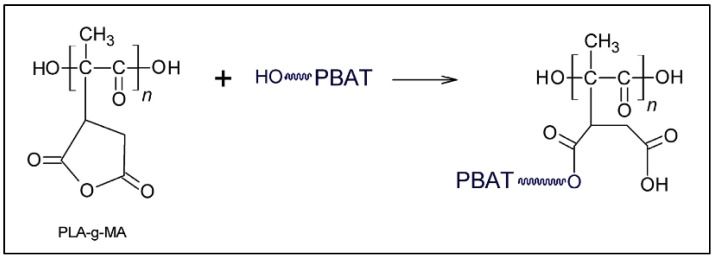
Possible reaction between PLA-*g*-MA and PBAT (from Rigolin et al. [50], copyright Elsevier, reproduced with permission).

Phetwarotai et al. studied PLA and PBS blends in the presence of PLA-*g*-MA or toluene diisocyante (TDI) as compatibilizers [53]. They showed that TDI is a more effective compatibilizer than PLA-*g*-MA for PLA/PBS blend films, their failure mode changed from brittle to ductile due to the improved compatibility. The effect of PLA-*g*-MA was also studied on PLA/PBAT/thermoplastic starch (TPS) ternary blend films. The thermal stability, tensile properties and compatibility of the PLA, PBAT, and TPS blends were slightly improved with the addition of the compatibilizer [54].

In the case of PLA/PCL with non-compatible blends, a reaction of MA with PCL is more likely to occur between the anhydride and OH groups on the PCL chain ends. As in the case of PBAT, it is the terminal alcohol of the PCL that reacts with the grafted anhydride of the PLA chain (Figure 12A) [55]. According to the authors, the compatibility effect of PLA-*g*-MA was shown by SEM micrographs (Figure 12B) and by a significant increase in the elongation at break of the material, from 7% in PLA or in a mixture of PLA and PCL to 53% in a compatibilized blend.

By reactive extrusion of the bio-based poly(glycerol succinate-co-maleate) (PGSMA) with PLA at 150–180 °C in the presence of a free radical initiator, a PLA-*g*-PGSMA graft copolymer was obtained that acts as an interfacial compatibilizer due to the double bond of the maleate unit in PGSMA [56]. The tensile strength of PLA/PGSMA blends was improved by almost 400% compared to that of pure PLA. This increase was caused by (i) the in situ formation of PLA-*g*-PGSMA graft copolymers and (ii) the crosslinking of PGSMA within the PLA matrix, which act as interfacial compatibilizers. Two-dimensional NMR and FTIR confirmed the formation of PLA-*g*-PGSMA, but the substitution degree on the PLA backbone was not evaluated. It is important to note that this work was performed with very-low-molecular-weight PGSMA (Mn < 1200 Da).

PLA-*g*-MA is also used to compatibilize PLA with thermoplastic polyurethane elastomers (TPU) and thermoplastic polyester elastomers (TPE). Charpy impact, toughness and fracture toughness of brittle PLA were improved when the blends were compatibilized by addition of PLA-*g*-MA without adversely damaging effects on other mechanical and thermal properties of the PLA blends [57].

##### Other PLA-Based Blends

PLA-*g*-MA serves as reactive compatibilizer for other polymer blends by reacting the anhydride functions on the second constituent of the blend. Examples include polyamides 11 and 12, where anhydride function reacts in the amine groups [58,59]. The ductility and impact strength of the compatibilized blends were increased by a factor two compared to non-compatibilized blends. A crosslinking agent (trimethylolpropane trimethacrylate) was also used to enhance the impact strength of the PLA-*g*-MA-containing blend. PLA/Polyamide6 (PA6) blends were compatibilized with PLA-*g*-IA (IA = itaconic anhydride), a compound similar to PLA-*g*-MA [60]. It was reported that IA can react with amine and amide functions of PA6. Unfortunately, a weak compatibility effect was obtained, likely due to the low concentration of IA moieties grafted onto PLA backbone (0.4 wt.%). PLA-*g*-IA is also a compatibilizer of PLA/Novatein (a protein-based thermoplastic) as shown in the SEM micrographs of Figure 13 [61].

Compatibilizing effects of PLA-*g*-MA on the properties of PLA/Soy Protein concentrate (SPC) blends were demonstrated by a 19% increase in tensile strength compared to the non-compatibilized blend [62]. Finally, PLA-*g*-MA also has the ability to provide some compatibility to PLA/mineral mixtures such as PLA/carbon nanotubes [22], PLA/Titanium oxide [63], PLA/halloysite [64], PLA/hydroxyapatite [65], PLA/talc [66], PLA polyhedral oligomeric silsesquioxane (POSS) [67].

#### 2.2.2. PLA-*g*-Glycidyl Methacrylate (PLA-*g*-GMA)

A second type of compound that allows for the reactive functionalization of the PLA chain is glycidyl methacrylate (GMA). Grafting is a free-radical reaction performed in a mixer (80 rpm) at 160 °C for 12 min in the presence of BPO [68]. A proposed scheme of the reaction is shown in Figure 14 [69].

The possibility of further functionalization is ensured by the presence of the new oxirane group on the grafted chain. As for PLA-*g*-MA, the main application lies in the compatibility of polymer blends. The graft content increased from 1.8 to 11.0 wt.% as the GMA concentration in the feed is varied from 5 to 20 wt.%. [68]. The degree of substitution, which is significantly higher than that of PLA-*g*-MA, is therefore the main advantage of GMA over MA. The characterization and properties of PLA-*g*-GMA (crystallization, characteristics, tensile stress, stress–strain curve, brittleness, thermal properties) compared to the one of PLA, are described by Kangwanwatthanasiri et al. [70].

The main graft copolymers obtained from PLA-*g*-GMA and their principal properties and applications are described in the following paragraphs. We found almost the same types of structures as those obtained with PLA-*g*-MA; therefore, they will not be described in detail.

##### PLA-*g*-Cellulosic Derivatives

PLA-*g*-GA is blended with cellulosic derivatives, fully renewable and degradable resources, to obtain a fully biodegradable product. For example, when used with starch, a small percentage of PLA-*g*-GMA plays the role of a compatibilizer, which is incorporated into both PLA and starch phases [68]. Essentially, when PLA-*g*-GMA is incorporated into PLA/cellulosic blends, the mechanical properties are improved: the starch tensile strength at break increased from 18.6 ± 3.8 to 29.3 ± 5.8 MPa, the tensile modulus from 510 ± 62 to 901 ± 62 MPa, and elongation at break from 1.8 ± 0.4 to 3.4 ± 0.6% [68].

Based on the reaction with PLA-*g*-GMA, the various PLA/cellulosic derivatives blends that were compatibilized are PLA–starch copolymers [68], PLA-treated arrowroot fiber [71], PLA–lignin [72,73] PLA–cassava pulp [74], PLA–cellulose [75], PLA–bamboo flour [76], PLA–rice straw fiber [77].

##### PLA-*g*-Polyesters

For the same reasons as described with MA, PLA/PBAT and PLA/PBS blends were compatibilized by the addition of PLA-*g*-GMA, with the main results of improved mechanical and thermal properties [69]. Specifically, the presence of PLA-*g*-GMA in a blend PLA/PBAT led to a decrease in crystallization rate, an increase in melt strength and viscosity, an improvement of tensile strength, and elongation at break, which are dependent on the proportion of PLA-*g*-GMA [78]. In a typical operating procedure, PLA, PBS or PBAT and PLA-*g*-GMA were melt blended together in a twin-screw extruder at a rotational speed of 30 rpm for several minutes. The temperature was between 150 and 170 °C [79]. The PLA/PBAT/PLA-*g*-GMA blends were successfully printed by 3D printing. A reaction mechanism between PBAT and PLA-*g*-GMA is described in Figure 15 [78]. With 10 wt.% of compatibilizer, the viscosity of the PBAT/PLA blend increased, and there was no longer a crystalline region of PBAT, showing an improved compatibility of PLA and PBAT.

PLA/cassava pulp/PBS ternary biocomposites were also compatibilized by PLA-*g*-GPA, with the mechanical properties of the PLA/cassava pulp/PBS composites being improved with the addition of PLA-*g*-GMA [80]. Similar to polyesters, PLA/thermoplastic polyurethane (TPU) blends were also compatibilized in the presence of PLA-*g*-GMA [81]. In this example, PLA-*g*-TPU acted as a compatibilizer for the blend PLA/TPU.

#### 2.2.3. PLA-*g*-Acrylic Acid (PLA-*g*-AA)

Another functionalization of the PLA chain is used in the field of the compatibilization of PLA-based blends, namely the grafting of acrylic acid (AA) to give a PLA-*g*-AA graft copolymer. Typically, AA grafting is performed under free radical conditions, by adding a mixture of AA and BPO to molten PLA in a mixer at 95 °C for a period of 6 h [82]. A PLA-*g*-PAA copolymer is formed, which can then react with alcohol functions of the cellulosic derivatives by esterifying the alcohol functions of the cellulosic compound (Figure 16).

As with PLA-*g*-MA, PLA-*g*-AA can compatibilize blends with natural cellulosic compounds, such as sisal fiber [31], wood flour [83], corn starch [84], rice husk [85], and hyaluronic acid [86]. In all cases, improvements in mechanical properties and/or biodegradation are obtained in the compatibilized mixtures. The compatibilization effect is shown by the size of the corn starch (CS) phase in PLA/CS and PLA/CS/PLA-*g*-corn starch. In a PLA/CS blend (50/50 *w*/*w*), the CS phase size decreased from 17.5 μm to 7.3 μm when PLA-*g*-corn starch was added in the blend [84].

Acrylic acid can also be graft-polymerized onto PLA chains in a solution using a photoinitiator, typically benzophenone, under UV irradiation at 254 nm [87]. Finally, the grafting of PAA onto the PLA backbone was also obtained by a free-radical reaction of BPO onto a solution of PLA in chloroform, followed by a reaction and polymerization of AA at 100 °C for 10 min under pressure. A drastic decrease in toughness and an increase in tensile modulus were observed in PLA-*g*-PAA as compared to PLA [88].

Inorganic–organic hybrid composites, based on mixtures of PLA and SiO_2_ [89] and TiO_2_ [90] generated via a sol–gel process, also showed improved mechanical and thermal properties when PLA was replaced by PLA-*g*-AA. This was attributed to stronger interfacial forces between carboxylic acid groups of PLA-*g*-AA and the residual Si-OH and Ti-OH groups [89].

#### 2.2.4. PLA-*g*-Halogen

The halogenation, in particular bromination, of the PLA chain is another method for reactive functionalization of the PLA chain. Usually, bromination is achieved by a free-radical mechanism. Typically, PLA is treated with N-bromo succinimide (NBS) in the presence of BPO over a period of 5 days [91]. There is no detectable chain degradation or crosslinking based on SEC results. Similarly, authors prepared chlorinated and iodinated PLAs. They used short-chain PLA (Mn = 2 kDa) to facilitate polymer characterization. The substitution degree depended on the halogen: from 3.2% for bromination and chlorination to 0.5% for iodination. However, the degree of bromination can be increased to 8% with microwave activation.

PLA-*g*-Br was used as a multisite macroinitiator for the ATRP of methyl methacrylate (MMA) and oligo ethylene glycol methacrylate (OEGMA) (Figure 17). Depending on the PLA-*g*-Br/MMA ratio, the side chains had different lengths. Complete bromine consumption was achieved during polymerization [92].

The bromination reaction was also performed on the surface of PLA films by treating the surface with NBS in H_2_O under UV irradiation. The incorporation of up to 3.7% bromine on the surface was achieved. Surface-initiated ATRP of quaternary ammonium methacrylate (QMA) chloride in the presence of CuBr and 2,2′-bipyridyl (bpy) was then performed, as shown in Figure 18. The cationic grafted surfaces are significantly more toxic to E.coli cells than genuine PLA, but no toxicity to HeLa cells upon contact was found [93].

A direct surface grafting (without any pre-halogenation) of poly(methacrylic acid) (PMAA) onto PLA was obtained after activation of the surface of a PLA film via photo-oxidation followed by the UV-induced polymerization of methacrylic acid. The grafting was confirmed in particular by FTIR analysis [94]. This method was employed to prepare a nano hydroxyapatite/g-PLA composite.

PLA nanofibers obtained by electrospinning were coated with PMMA by plasma polymerization [95]. The coated PLA fibers showed an increase in diameter from 250 nm to 700 nm. MTT assays and cells count showed that the PLA-*g*-PMMA copolymers form intrinsically biocompatible systems.

#### 2.2.5. Other PLA-Based Graft Copolymers

##### PLA-*g*-Nitrilotriacetic Acid (PLA-*g*-NTA)

In the field of “green packaging”, it is desirable to have non-migratory, metal-chelating and biodegradable materials. To this end, metal-chelated nitrilotriacetic acid (NTA) was grafted onto PLA using a classical radical mechanism, as shown in Figure 19 [96]. The grafting was evidenced by ATR-FTIR and XPS. Significant radical scavenging and metal-chelating efficacies as well as the ability to delay the degradation of ascorbic acid showed the antioxidant capacity of PLA-*g*-NTA. However, this NTA grafting was not followed by any polymer grafting, even if the grafted carboxylic acids are functional groups.

##### PLA-*g*-Vinyltrimethoxysilane (PLA-*g*-VTMS)

PLA-*g*-VTMS was prepared by a free radical reaction of vinyltrimethoxysilane **(**VTMS) on PLA in the presence of DCP in a Brabender at 190 °C for 5 min. The trimethoxysilane was then hydrolyzed to allow the crosslinking of the PLA chains to improve the mechanical properties of the PLA nanofibers (Figure 20) [97]. Electrospun nanofibrous mats based on PLA/NCC and PLA-*g*-silane/NCC nanocomposites were fabricated and compared. PLA-*g*-VTMS was used to improve the mechanical properties of PLA/nanocrystalline cellulose (NCC). In particular, the impact of NCC on improving tensile strength was notable, even though no chemical reaction of PLA-*g*-VTMS with NCC is reported. In addition, a cytotoxicity assessment showed the biocompatibility of the modified nanofibers, making them good candidates for tissue engineering applications.

##### PLA-*g*-Poly(Vinyl Pyrrolidone) (PLA-*g*-PVP)

A PLA film was treated with a solution of N-vinyl pyrrolidone (NVP) in methanol and AgNO_3_ using ^60^Co γ−radiation polymerization at a dose of 1–30 kGy at room temperature. After washings, a PLA-*g*-PVP film was formed on the surface. Silver nanoparticles were also immobilized on the film surface. A surface grafting ratio, in the range of 25–49%, is assessed by the FTIR ratio of the bands at 1660 cm^−1^ of PVP and the sum of the bands at 1660 cm^−1^ and 1750 cm^−1^ of PLA [98]. There is no indication of the grafting degree of PVP onto the PLA backbone. It is noted, however, that PVP grafting significantly accelerated PLA degradation and does not impede cell proliferation [99]. Controlled variation in the grafting ratio could broaden the applications of this material in tissue engineering scaffolds, drug delivery, and the prevention of post-surgical adhesion.

#### 2.2.6. Anionic Derivatization

The anionic derivatization of PCL was described by Ponsart et al. [19]. It is a remarkably powerful one-pot two-step method for grafting many types of substituents on the PCL backbone. This method is theoretically applicable to many polyesters, but it leads to varying degrees of chain cleavage depending on the nature of the polyester, due to the anionic medium caused by the presence of lithium diisopropylamide. Even though PCL is relatively resistant to this basic medium, which is not the case with PLA, There are still many chain breakings. Nevertheless, El Habnouni et al. applied the method to the surface of the PLA film and nanofibers in a non-solvent medium that causes only moderate-chain scissions and allows for the preparation of functional PLA surfaces [100]. In particular, this approach was exploited with propargylated PLA allowing the grafting of bioactive polymers through CuAAC or thiol-yne click reactions. Therefore, anti-biofilm and bactericidal PLA surfaces were obtained by the reaction of α-azido QPDMAEMA (quaternized poly(2-(dimethylamino)ethyl methacrylate)) or thiol-functional polyaspartamide derivatives (Figure 21) [101,102].

The main copolymers grafted onto the PLA chain, precursors, copolymers and literature references are summarized in Table 3.

## 3. Conclusions

Research on the chemical modifications of the PLA backbone to yield PLA-*g*-polymer graft copolymers is scarce. These modifications mostly occur via a radical mechanism in the presence of a peroxide, leading to the covalent substitution of a reagent on the methine proton of the PLA chain. These reactions are essentially carried out in mass at high temperature in a mixer or an extruder. The main substituents are anhydride or epoxy groups that allow a reactive compatibilization of PLA-based polymer blends. The degree of substitution remains low (<2%) but allows for significant improvements in properties, mainly in mechanical properties. The reactions of the anhydride or epoxide functions grafted on the PLA chain with other polymers (cellulose derivatives, polyesters, polyamides, natural gums, PMMA) lead to the formation of numerous graft copolymers whose backbone is PLA. If these PLA-*g*-polymers are mostly described for the compatibilization of PLA-containing blends, it appears that the applications of these PLA-based grafted copolymers could cross over into biomedical and environmental fields, because they are intrinsically biocompatible systems. In any case, the low degree of grafting obtained in these grafting reactions highlights the importance of finding new grafting approaches to develop functionalization on the PLA chain in order to obtain new PLA-based graft copolymers.

## Figures and Tables

**Figure 1 molecules-27-04135-f001:**
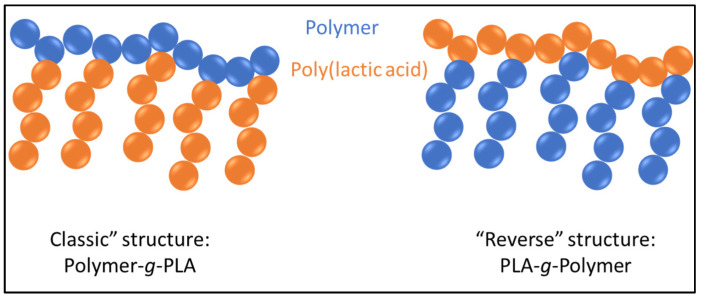
Illustration of “classic” and “reverse” structures of PLA-based graft copolymers.

**Figure 2 molecules-27-04135-f002:**
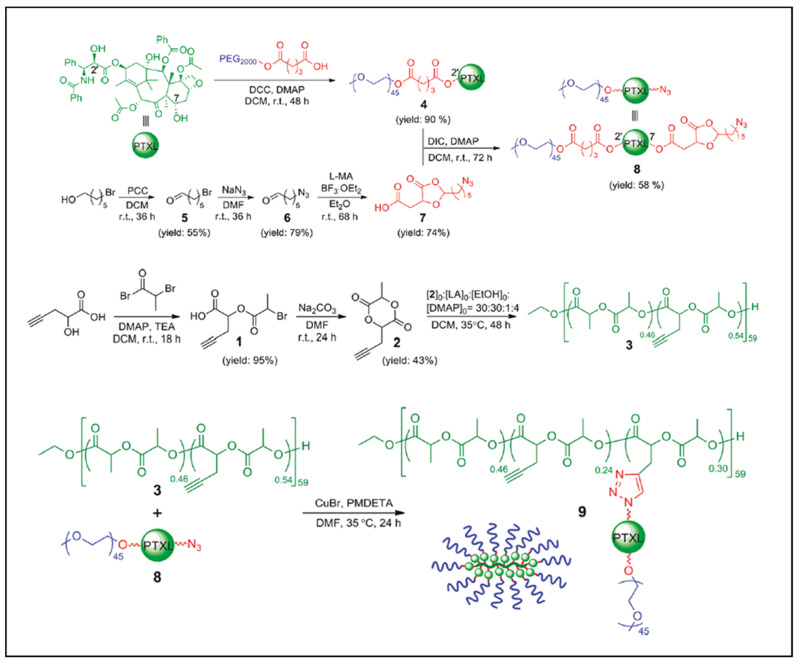
Reaction scheme for the synthesis of PLA-*g*-Paclitaxel-PEG (adapted from Yu et al. [16], Copyright American Chemical Society, reproduced with permission).

**Figure 3 molecules-27-04135-f003:**
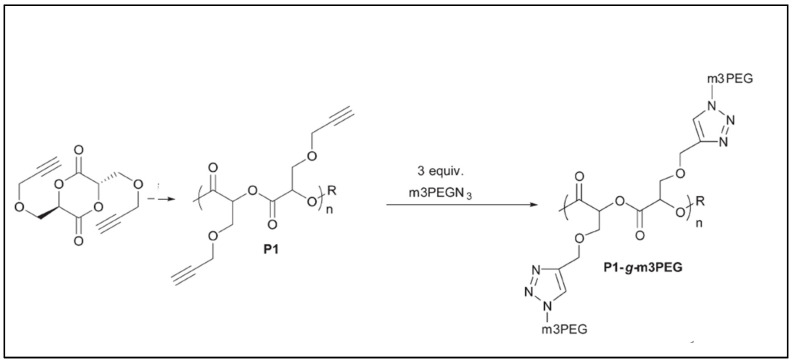
Reaction scheme for the synthesis of dipropargyloxymethyl PLA and the grafting of PEG via click a reaction (adapted from Zhang et al. [17], Copyright Royal Society of Chemistry, reproduced with permission).

**Figure 4 molecules-27-04135-f004:**
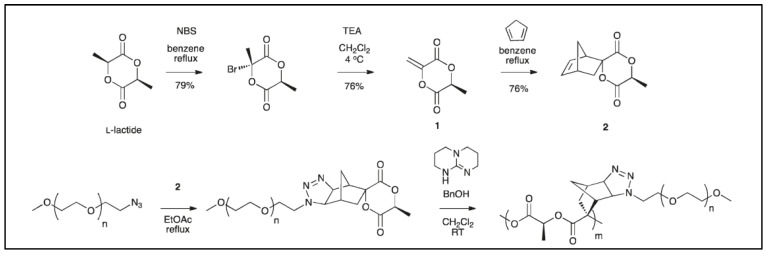
Reaction scheme for the preparation of PLA-*g*-PEG (from Castillo et al. [18], copyright American Chemical Society, reproduced with permission).

**Figure 5 molecules-27-04135-f005:**
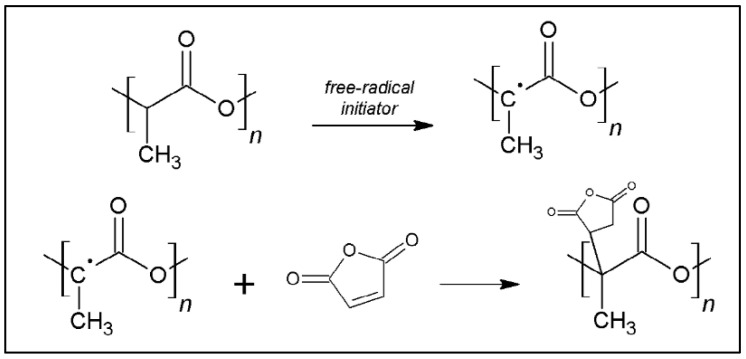
Reaction scheme for the grafting of MA on PLA (from González-López et al. [21], copyright Taylor and Francis, reproduced with permission).

**Figure 6 molecules-27-04135-f006:**
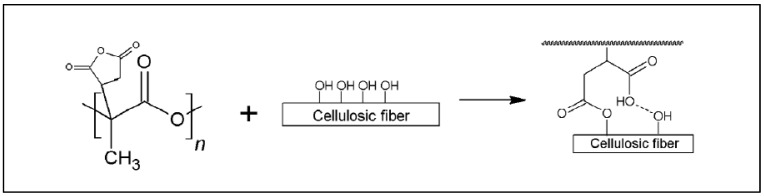
Reaction scheme for the grafting of a cellulosic fiber on PLA-*g*-MA (from González-López et al. [21], copyright Taylor and Francis, reproduced with permission).

**Figure 7 molecules-27-04135-f007:**
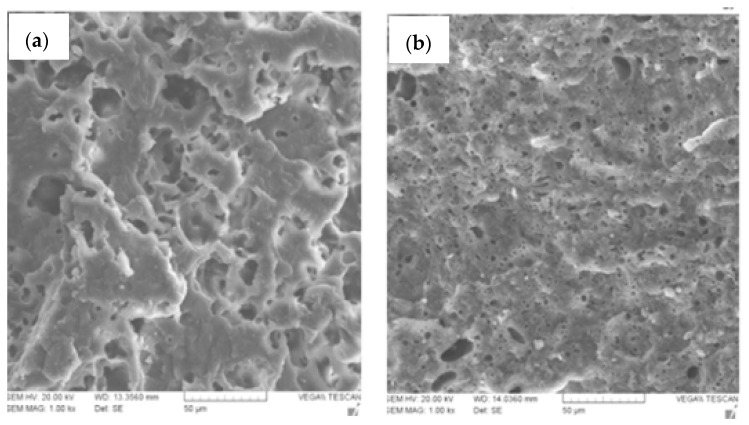
SEM micrographs of PLA/TPS blends (70/30 *w*/*w*) (**a**) without and (**b**) with 2 phr PLA-*g*-MA (adapted from Moghaddam et al. [39], copyright Springer Science, reproduced with permission).

**Figure 8 molecules-27-04135-f008:**
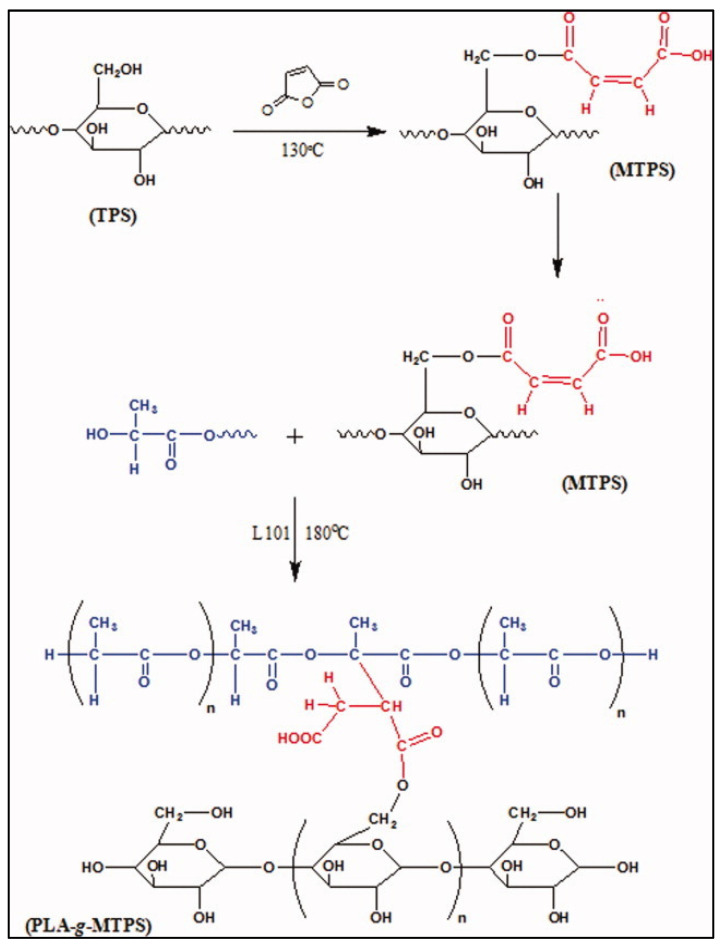
Reaction scheme of the MTPS formation and coupling to PLA (from Wootthikanokkhan et al. [42], copyright Wiley-VCH GmbH. Reproduced with permission).

**Figure 9 molecules-27-04135-f009:**
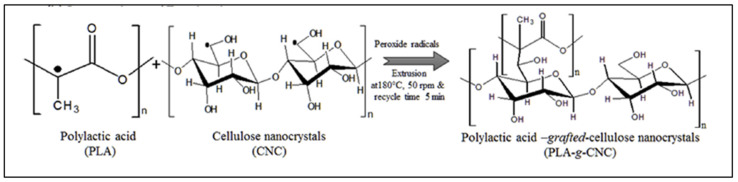
Reaction scheme for the synthesis of PLA-*g*-CNC (from Dhar et al. [43], copyright Elsevier, reproduced with permission).

**Figure 10 molecules-27-04135-f010:**
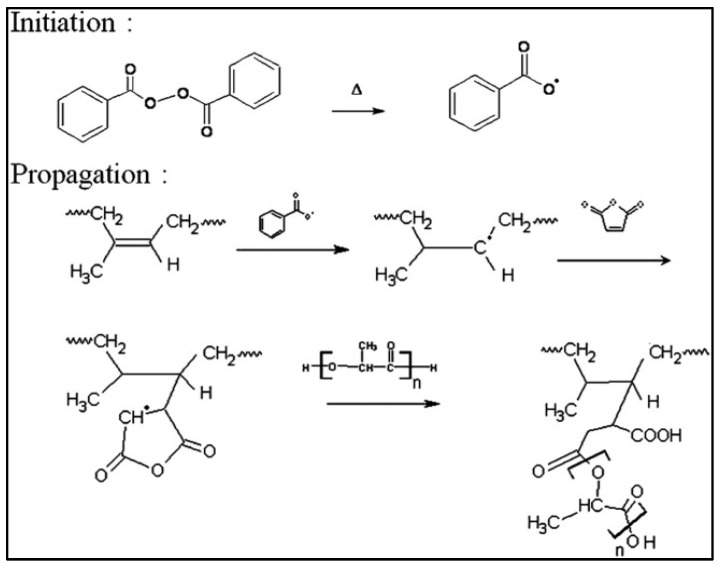
Proposed mechanism for the grafting of NR-MA on PLA (from Thepthawat et al. [47], copyright Wiley-VCH GmbH. Reproduced with permission).

**Figure 12 molecules-27-04135-f012:**
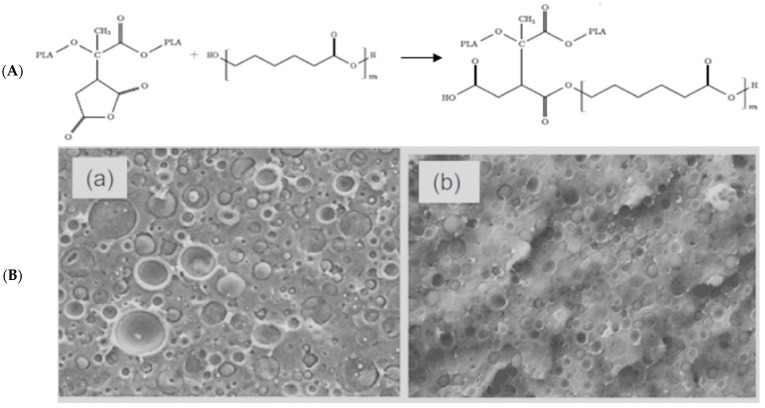
(**A**) Proposed reaction scheme between PLA-*g*-MA and PCL (**B**) SEM micrographs of PLA/PCL blends (**a**) without and (**b**) with 10% PLA-*g*-MA (adapted from Gardella et al. [55], copyright Springer, reproduced with permission).

**Figure 13 molecules-27-04135-f013:**
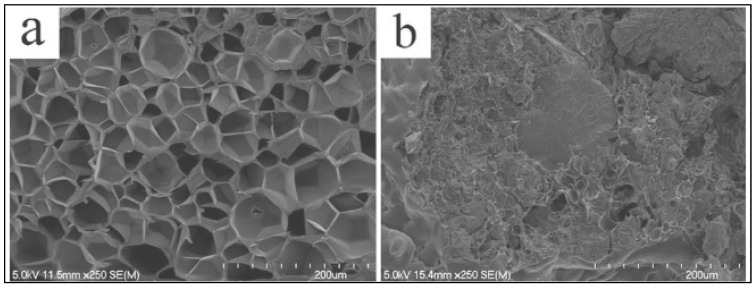
SEM micrographs of PLA/novatein 90/10 blends (**a**) uncompatibilized, (**b**) compatibilized (adapted from Walallavita et al. [61], copyright Wiley-VCH GmbH. Reproduced with permission).

**Figure 14 molecules-27-04135-f014:**
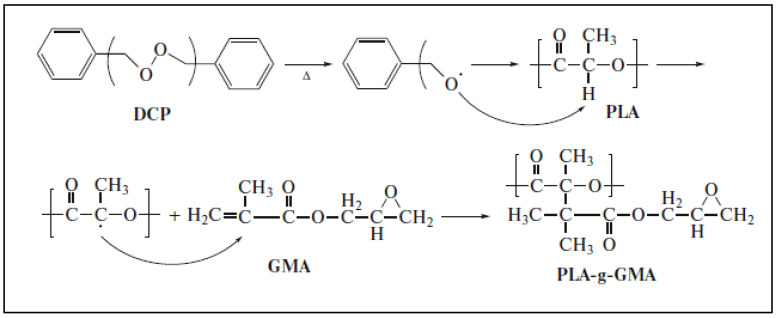
Reaction scheme for the grafting of GMA on PLA backbone (from Gu et al. [69], copyright Springer Nature, reproduced with permission).

**Figure 15 molecules-27-04135-f015:**
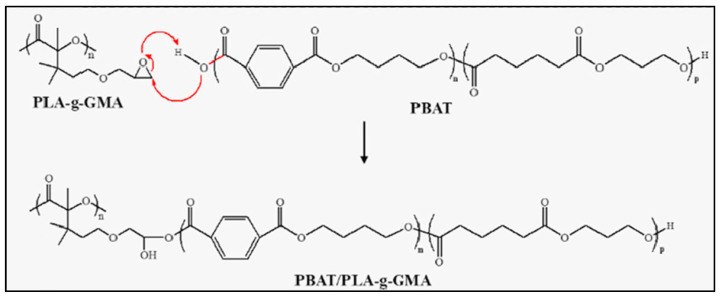
Mechanism of the reaction between PLA-*g*-GPA and PBAT (from Lyu et al. [78], copyright Elsevier, reproduced with permission).

**Figure 16 molecules-27-04135-f016:**
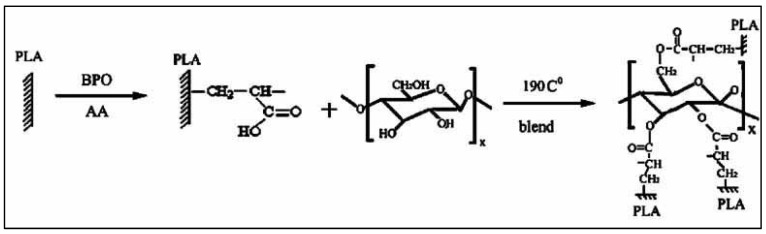
Reaction scheme for the grafting of acrylic acid on PLA and reaction with starch (from Wu, [82], copyright Wiley-VCH GmbH. Reproduced with permission).

**Figure 17 molecules-27-04135-f017:**
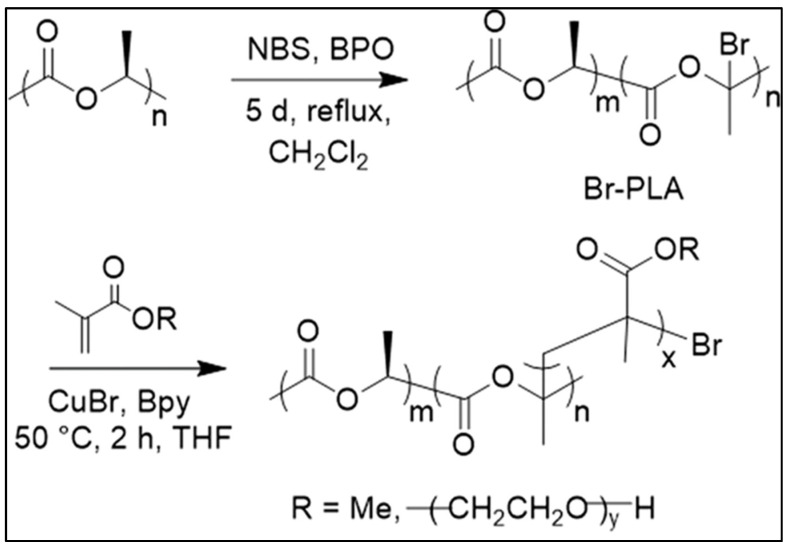
Formation of PLA-*g*-PMMA and PLA-*g*-POEGMA by ATRP (from Kalelkar et al. [92], copyright American Chemical Society, reproduced with permission).

**Figure 18 molecules-27-04135-f018:**
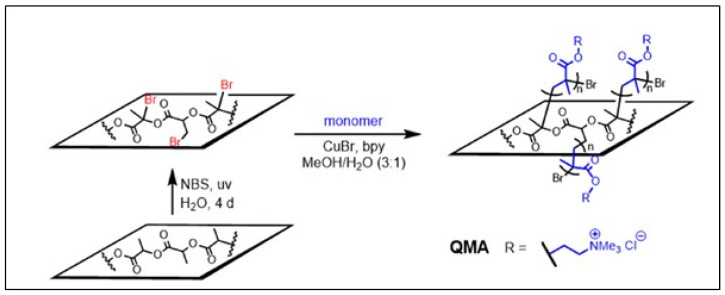
Surface-initiated ATRP of methacrylate monomers (from Kalelkar et al. [93], copyright Elsevier, reproduced with permission).

**Figure 19 molecules-27-04135-f019:**
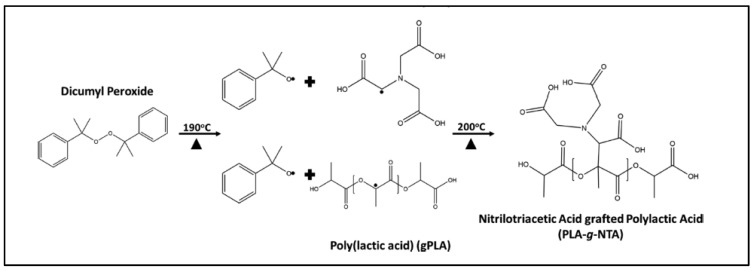
Reaction scheme for the grafting of NTA on PLA (from Herskovitz et al. [96], copyright American Chemical Society, reproduced with permission).

**Figure 20 molecules-27-04135-f020:**
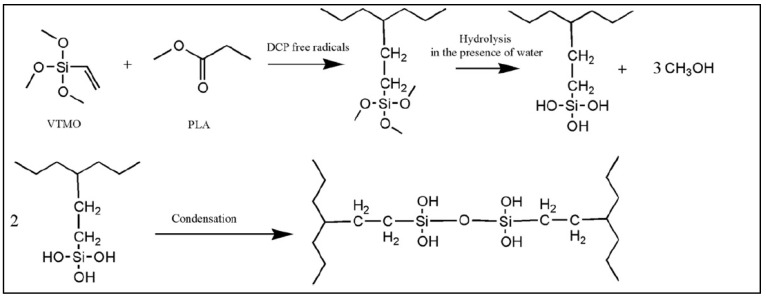
Synthesis scheme of PLA-*g*-VTMS, hydrolysis of methoxysilane, and crosslinking of PLA chains (adapted from Rahmat et al. [97], copyright Elsevier, reproduced with permission).

**Figure 21 molecules-27-04135-f021:**
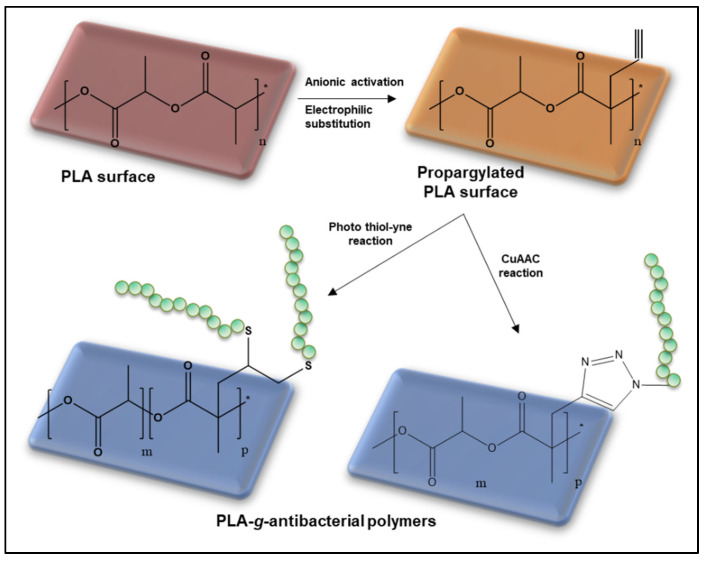
General synthesis scheme of PLA-*g*-antibacterial polymers from propargylated PLA surface (adapted from Sardo et al. [102] and El Habnouni et al. [101]).

**Table 1 molecules-27-04135-t001:** Variations in molecular weight and grafting percentage during functionalization of PLA by MA (extract of data from Muenprasat et al. [23]).

PLA Molecular Weight	PLA-*g*-MA Molecular Weight	Grafting %
294,000	196,000	0.65
92,000	76,000	1.25

**Table 2 molecules-27-04135-t002:** Mechanical properties of PLA, PLA/CNF and PLA/CNF/PLA-*g*-MA (extract of data from Ghasemi et al. [24]).

	Tensile Strength (MPa)	Strain at Break (%)	Tensile Modulus (GPa)
Original PLA	22.4	2.3	1.1
PLA/CNF(5%)	34.8	3.0	1.3
PLA/CNF(5%)/PLA-*g*-MA (5%)	60.3	5.5	1.5

**Table 3 molecules-27-04135-t003:** Main PLA-based grafted copolymers according to literature.

Precursor: Functionalized PLA	PLA-*g*-Copolymer	Refs.
PLA-*g*-MA	cellulosic derivatives	[24,40,41]
	luffa	[26]
	flax	[27]
	coffee grounds	[103]
	wood flour	[29,30]
	rice husk	[29,30]
	sisal	[31]
	straw	[32]
	bamboo	[33]
	cassava	[34,35]
	starch	[36,39,42]
	lemongrass	[37]
	natural rubber	[44,45]
	polyesters	
	PBAT	[48,49,50]
	PBAT/TiO_2_	[51]
	PBAT/CaCO_3_	[52]
	PBAT/starch	[54]
	PBS	[53]
	PCL	[55]
	PGSMA	[56]
	TPU	[57]
	PA	[58,59]
	soy protein	[62]
	mineral compounds	[22,63,64,65,66,67]
PLA-*g*-IA	polyamide	[60,61]
PLA-*g*-GMA	cellulosic derivatives	
	starch	[68]
	arrowroot	[71]
	lignin	[72,73]
	cassava	[74]
	cellulose	[75]
	bamboo	[76]
	rice-straw	[77]
	polyesters	
	PBAT	[69,78,79]
	PBAT/cassava	[80]
	TPU	[81]
PLA-*g*-AA	cellulosic derivatives	
	starch	[82,84,90]
	sisal	[31]
	wood flour	[83]
	rice husk	[85]
	mineral compounds	[89]
	hyaluronic acid	[86]
PLA-*g*-halogen	PMMA and POEGMA	[92,93,94]
	PVP	[98,99]
PLA-*g*-alkyne	PEG	[17]
Direct grafting	cellulose nanocrystals	[43]
	natural rubber	[46,47]
	PMMA	[95]
Surface-anionic derivatization	QPDMAEMA	[101]
	α,β-poly(N-2-hydroxyethyl)-D,L-aspartamide	[102]

## Data Availability

Not applicable.

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
