# Peer review of "Poly(Lactic Acid)-Based Graft Copolymers: Syntheses Strategies and Improvement of Properties for Biomedical and Environmentally Friendly Applications: A Review"

_molecules, 2022, doi:10.3390/molecules27134135_

Round 1
Reviewer 1 Report
Review of manuscript entitled “Poly (lactic acid)-based graft copolymers: syntheses strategies and improvement of properties for biomedical and environmental fields. A review”. J. Coudane, H. Van Den Berghe, J. Mouton, X. Garric and B. Nottelet.
This review addresses the recent advances in synthesis of PLA-based graft copolymers that can be used as blend comptabilizers. Specifically the paper is focused on the synthesis of copolymers where PLA is the main chain. Two synthesis methods are described: copolymerization of lactide with a functionalized lactide and direct chemical modification of a preformed PLA chain.
The topic of the manuscript is interesting and fits well the scope of the journal. Also, it is well written, well structured and the references presented are actual and representative of the subject of the manuscript. The only thing to mention is that, in my opinion, the title should be modified. The part “biomedical and environmental fields” should be changed for “biomedical and enviromentally friendly applications” or something similar. Apart from this detail I think the manuscript can be published as it is.
Author Response
Dear reviewer,
I thank you for the work in reviewing my manuscript. I agree with the change in the title of the review and have corrected it in the amended manuscript.
Best regards
Jean Coudane

Reviewer 2 Report
The manuscript under consideration constitutes the review of PLA-based graft copolymers. It would be suitable for publication after revision.
In the manuscript, the word biocompatibility is repeated 6 times. However, biocompatibility is not a material property but a material-host system. Therefore, the term “biocompatibility” reflects the ability of a material to perform with an appropriate host response in a specific application. According to Prof. David F. Williams, there is no such thing as a biocompatible material: http://dx.doi.org/10.1016/j.biomaterials.2014.08.035.
I would suggest reworking.
Author Response
Dear reviewer,
I agree with you - and David F. Williams - that the term "biocompatible material" is often inappropriate, even misleading. Biocompatibility depends on the material-living environment system under consideration.
However, the use of the expression "biocompatible material" is extremely common, even in recent high-level scientific publications! Maybe the expression "intrinsically biocompatible system" would be more appropriate.
In order to be more correct in the use of the word biocompatibility, we have changed the expression "biocompatible material" whenever possible.
best regards
Jean Coudane

Round 2
Reviewer 2 Report
The revised manuscript is suitable for publication in my opinion.